# Enhancing Accessible Communication:
# from European Portuguese to Portuguese Sign Language

**Catarina Sousa** and **Luísa Coheur**
INESC-ID, Lisbon, Portugal
Instituto Superior Técnico, University of Lisbon, Portugal
catarinasousa2000@tecnico.ulisboa.pt
luisa.coheur@tecnico.ulisboa.pt

**Mara Moita**
Universidade Católica Portuguesa, Portugal
maramoita@ucp.pt

## Abstract

Portuguese Sign Language (LGP) is the official language in deaf education in Portugal. Current approaches in developing a translation system between European Portuguese and LGP rely on hand-crafted rules. In this paper, we present a fully automatic corpora-driven rule-based machine translation system between European Portuguese and LGP glosses, and also two neural machine translation models. We also contribute with the LGP-5-Domain corpus, composed of five different text domains, built with the help of our rule-based system, and used to train the neural models. In addition, we provide a gold collection, annotated by LGP experts, that can be used for future evaluations. Compared with the only similar available translation system, PE2LGP, results are always improved with the new rule-based model, which competes for the highest scores with one of the neural models.

## 1 Introduction

According to the Portuguese Association of Deaf people[1], there are around thirty thousand deaf people that use Portuguese Sign Language (LGP) in their daily lives. However, European Portuguese (EP) and LGP are two different linguistic systems, and communication between hearing and hearing-impaired people is difficult, leading to a communication gap between both groups. There have been some attempts at developing a system that translates EP into LGP glosses (Almeida, 2014; Gaspar, 2015; Escudeiro et al., 2015; Ferreira, 2018), but the majority only present toy examples, relying on a small set of hand-crafted rules and disregarding non-manual movements. PE2LGP (Gonçalves et al., 2021; Lacerda et al., 2023) is a rule-based

translation system from EP to LGP. Some of its rules are hand-crafted and some are automatically extracted from a linguistic corpus with annotations of LGP videos[2], from now on COLIN. COLIN is the only existing LGP annotated corpus (to the best of our knowledge). It currently consists of 113 hours of video recordings, with 20 of them being annotated at various linguistic levels (ELAN[3] (Sloetjes and Wittenburg, 2008) was used). The videos in the corpus were recorded between 1992 and 2019 and feature hearing-impaired signers ranging from 4 to 89 years old. PE2LGP still has some limitations, as the subset of COLIN used to develop its rules is very small (three minutes) and it is necessary to perform some manual tasks to create PE2LGP translator's grammar. In this paper, we take advantage of an extended version of the corpus used in PE2LGP and present a rule-based system that is now fully automatic. We use this rule-based system to create a parallel corpus between EP and LGP, from now on LGP-5-Domain (LGP5), with text from different domains – simple sentences, social media, poetry, dialogue, and news. LGP5 is then used to fine-tune two large multilingual neural machine translation models, and evaluated on a gold collection, built by LGP experts[4]. With this work, we hope to contribute to benchmarking this task.

To illustrate the task at hand, Table 1 presents examples of EP sentences and their corresponding translation into LGP glosses[5]. The first row con-

---

[1] https://apsurdos.org.pt/

[2] Developed in the project "Corpus Linguístico e AVATAR da Língua Gestual Portuguesa (PTDC/LLT-LIN/29887/2017)."

[3] https://archive.mpi.nl/tla/elan

[4] Code and datasets available at https://github.com/lcoheur/EP2LGP5.0

[5] We use the same notation as in PE2LGP.

| EP | Glosses in LGP |
|---|---|
| A Ana gosta de massa? | {DT(A-N-A) GOSTAR}(q) MASSA |
| (Ana likes pasta?) | ({(DT(A-N-A) PASTA LIKE)}(q)) |
| A rainha foi à praia. | MULHER REI PRAIA IR |
| (The queen went to the beach.) | (WOMAN KING BEACH GO) |

Table 1: Examples of translations EP/LGP.

tains an interrogative sentence, marked by { }(q) – facial expression denoting a question that involves raising the chin, tilting the head back, and frowning. Since "Ana" is a proper name, it should be finger-spelled – represented by DT(A-N-A). In the declarative sentence of the second row "rainha" (queen) is translated to "MULHER REI" (WOMAN KING).

## 2 Related Work

We will only consider the translation *to* a sign language and not *from* a sign language, which is out of the scope of this paper. Early automatic translation systems, such as the ones discussed in (San-Segundo et al., 2008; Zhao et al., 2000; Brour and Benabbou, 2019), adopted a rule-based approach for performing the translation between the source and the target languages. An example of an (expert-defined) rule employed in the *ATLASLang MTS* system (Brour and Benabbou, 2019) is shown in Equation 1.

$$If \; Gender(w_i) = feminine$$
$$Then \; t_i = w_i + (female). \quad (1)$$

Equation 1 checks if a word has a feminine gender and adds the term *female* after it. Notice that the involved types of grammar could vary. For instance, the work described in (Zhao et al., 2000) uses a Lexicalized Tree Adjoining Grammar to translate English words into American Sign Language glosses.

Regarding LGP, there are a few attempts at creating a translation system such as the ones described in (Escudeiro et al., 2015; Oliveira et al., 2019; Gaspar, 2015). VIRTUALSIGN (Escudeiro et al., 2015; Oliveira et al., 2019) is a system that performs bidirectional translation between Portuguese and LGP. Regarding the text-to-sign translation, the input sentence is passed through a set of grammar rules and, afterward, the system directly associates each word in the sentence with a corresponding

sign stored in a database. This database contains the needed information to animate a 3D avatar. Unfortunately, no information is available regarding the creation of the translation rules and the evaluation of this system.

PE2LGP has a rule-based translation system (Gonçalves et al., 2021), an avatar to perform LGP, and a database with signs (Cabral et al., 2020; Lacerda et al., 2023). Its rules were semi-automatically extracted from COLIN. In this work, we take advantage of the current corpus and we also automate the whole process.

Another work that should be mentioned is IF2LGP (Gaspar, 2015) which consists of two modules. The first is responsible for conducting syntactic and morphological analysis, while the second contains translation rules to convert Portuguese words into LGP glosses. However, the creation of these translation rules was based on a small dataset of ten sentences.

Over the past years, significant progress has been made in the field of sign language translation, thanks to the advancements in statistical and neural machine translation. For instance, the approach proposed by San-Segundo et al. (2008) uses a phrase-based method, trained using parallel corpora; also, the *ATLASLang NMT* system (Brour and Benabbou, 2021) employs a neural machine translation approach.

Regarding LGP, a neural approach is also explored by Alves et al. (2022). It adopts a hybrid structure combining rule-based and neural machine translation approaches. According to the authors, the dataset consists of 150,000 sentences. However, the grammar and the data are not available.

## 3 Towards the Portuguese Sign Language

### 3.1 The LGP Corpus

We were given access to COLIN, and used the forty-five minutes of the corpus that had all the needed syntactic annotations to reconstruct the expert translator's grammatical rules.

### 3.2 Improving the Rule-based Model

The corpora-driven rule-based approach used in this translation system is depicted in Figure 1.

The first module automatically extracts linguistic information from the corpus, generating translation rules and a bilingual dictionary between EP and LGP. To do so, we extract from ELAN the Portuguese sentences, the LGP corresponding transla-

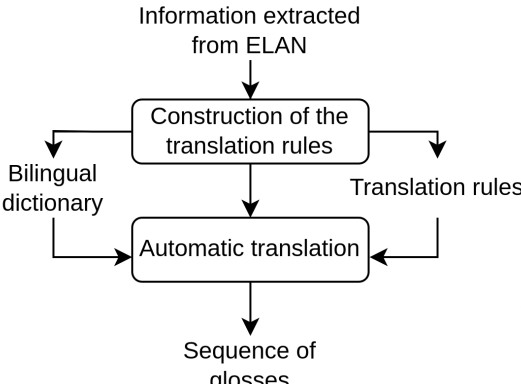

Figure 1: Pipeline of our rule-based approach.

tion in glosses, and their grammatical information – part-of-speech tags, subjects, and objects. Having the information regarding the LGP sentences, we analyze the EP ones to also obtain their grammatical information. Having information from both languages, the alignment between the Portuguese words and the LGP glosses is performed. This alignment is based on an algorithm of similarity measures, string matching, and semantic similarity. From the aligned word-sign pairs, the rules and the bilingual dictionary are created. The second module uses these translation rules and the bilingual dictionary to translate EP into LGP, where the LGP sentence is represented by a sequence of glosses with markers indicating facial expressions and fingerspelled words. When an EP sentence enters the system, the sentence is analyzed and its structure is kept. Then, the distance between the Portuguese structure and the structure of the system's rules is calculated. The rule with the lower distance is the most similar one to the original sentence and is applied to convert the Portuguese structure into the LGP one.

PE2LGP has 61 *general syntactic rules*, some hand-crafted. Our current proposal handles 238 of such rules (218 are for declarative sentences, 7 for negative ones, and 13 for interrogatives). Eq. 2 shows an example that states that if the Portuguese sentence has the canonical order *Verbal Phrase (VP) – Noun Phrase (NP)*, then, the LGP sentence will have the canonical order *NP – VP*.

$$VP \ NP \ \rightarrow \ NP \ VP \quad (2)$$

PE2LGP has 90 *morphosyntactic rules*, and our proposal adds 228 of such rules. As an example, Eq. 3, states that, a Portuguese sentence with the syntactic structure *Verb – Adjective – Noun – Adjec-*

*tive*, will be translated into an LGP sentence with the syntactic structure *Verb – Noun – Noun*.

$$V1 \ ADJ1 \ N1 \ ADJ2 \ \rightarrow \ V1 \ N2 \ N1 \quad (3)$$

This rule can be applied, for instance, to the sentence:

$$Há \ grandes \ desenvolvimentos \ artísticos.$$
$$(There \ are \ great \ artistic \ developments.)$$
$$(4)$$

The sentence presented in Eq 4 is translated into:

$$TER - MUITO \ ARTE \ DESENVOLVIMENTO$$
$$(HAVE - A - LOT \ ART \ DEVELOPMENT)$$
$$(5)$$

This translation occurs because there is an entry in the bilingual dictionary that is:

$$Haver \ grande \ \rightarrow \ TER - MUITO$$
$$(There \ is \ great \ \rightarrow \ HAVE - A - LOT)$$
$$(6)$$

Additionally, as the rule presented in Eq. 3 suggests, the first adjective can now be removed – as it was merged with the first verb – the noun in the original sentence is moved to the last word in the translation sentence and the second adjective is converted into a noun – its lemma – and it is positioned in the middle of the sentence. The alignment between the rules and the source and target sentence is depicted in Fig. 2.

(There are (V1)  great (ADJ1)  artistic (ADJ2)  developments. (N1))

Há (V1) grandes (ADJ1) desenvolvimentos (N1) artísticos. (ADJ2)

TER-MUITO (V1)      ARTE (N2)      DESENVOLVIMENTO (N1)

(HAVE-A-LOT  (V1)      ART  (N2)      DEVELOPMENT  (N1))

Figure 2: Alignment between the source sentence (Eq. 4) and the target sentence (Eq. 5) using the rule depicted in Eq. 3

### 3.3 The Neural Models

We tuned two multilingual models: the mBART model (Liu et al., 2020) and the M2M model (Fan et al., 2021). Given that the LGP glosses are written in Portuguese, both the input and output languages were set to Portuguese. As a result, a Portuguese-to-Portuguese translator was created, and fine-tuning

was performed to allow the models to learn how to translate to LGP glosses. The base models and the associated weights are from the HuggingFace's Transformers package (Wolf et al., 2020). For the mBART model, we used the *mbart-large-50-many-to-many-mmt* checkpoint and, for the M2M model, we used the *m2m100_1.2B* checkpoint. The strategy implemented to fine-tune these models was similar for both as we used the default hyper-parameters except for the batch size and the number of epochs. We used a batch size of 2 for both models since it was the maximum feasible given our computational resources. We ran our model for 3 epochs since it would start to overfit if we increased the number. The employed fine-tuning data consisted of the parallel corpus created with the rule-based approach, that is the LGP5 corpus, described next.

## 4  The LGP5 Dataset

In the following, we describe the LGP5 dataset that comprises 37,500 automatically annotated sentences from 5 different domains (7,500 from each domain) that were used to train the neural models. Additionally, 200 sentences were manually annotated (gold collection), 40 from each domain.

### 4.1  Gathering Data

In order to have a rich collection of **simple sentences**, we extracted Portuguese sentences from Tatoeba[6]. With the aim to expose the models to the unique linguistic characteristics prevailing in online social interactions – slang, abbreviations, and other aspects of contemporary communication commonly found on **social media** platforms – we used a dataset with Portuguese tweets from Kaggle[7]. **Poetry** texts – the complete literary work written by Fernando Pessoa, a famous Portuguese poet – were also used to enable training with a broader range of sentence structures. Kaggle[8] was, again, our source of data. We also considered **dialogues** to obtain sentences from the everyday speech of Portuguese people. For this we used the dataset described in (Csaky and Recski, 2021). Finally, the last dataset was composed of **news** articles[9]. Training the model with sentences from this do-

main not only exposes the model to an organized and coherent language but also to a broad range of topics.

### 4.2  The LGP5 Parallel Corpus

Our rule-based model was used to translate the gathered corpus. As a result, a new dataset comprising 37,500 EP/LGP pairs was generated (examples can be seen in Table 4, in Appendix A).

### 4.3  The Gold Collection

As previously said, the gold collection consists of 200 pairs of sentences EP/LGP – 40 sentences from each of the five domains. These sentences were annotated by the rule-based system and given to two LGP experts. Each one validated/corrected 100 sentences.

## 5  Experiments

For the evaluation experiments, we evaluated our systems with three metrics: BLEU-4 (Papineni et al., 2002), RougeL (Lin, 2004), and TER (Snover et al., 2006).

In a preliminary experiment, we evaluated our models in a test set described in (Gonçalves et al., 2021). This test set has 58 sentences EP/LGP. We also evaluated our models against PE2LGP. Table 2 shows the results. Our rule-based approach (RB) is better than PE2LGP, and M2M is the best model overall.

| | PE2LGP | RB | mBART | M2M |
|---|---|---|---|---|
| BLEU ↑ | 59.29 | 62.16 | 63.93 | **68.55** |
| RougeL ↑ | 76.91 | 79.56 | 79.79 | **82.50** |
| TER ↓ | 84.31 | 74.45 | 75.55 | **62.41** |

Table 2: Preliminary results.

Next, we used our gold collection to evaluate the same models (Table 3).

M2M is not always the one that obtains the highest scores, as our rule-based system is the best system in the social media dataset, for all the measures, and in some of the other domains, for some of the metrics. PE2LGP initially seemed to have the best *TER* score in the poetry domain. However, a closer look revealed that it failed to provide a translation for four out of the forty sentences. This affected the *TER* calculation, making the final scores appear higher than they actually were. Considering this, PE2LGP's scores were consistently

---

[6]https://tatoeba.org/en
[7]https://www.kaggle.com/datasets/augustop/portuguese-tweets-for-sentiment-analysis
[8]https://www.kaggle.com/datasets/luisroque/the-complete-literary-works-of-fernando-pessoa
[9]dados.gov.pt/en/datasets/noticias

|  | PE2LGP | RB | mBART | M2M |
|---|---|---|---|---|
| **Simple sentences** | | | | |
| BLEU ↑ | 37.93 | 41.37 | 41.53 | **43.01** |
| RougeL ↑ | 59.57 | 63.83 | 64.15 | **67.77** |
| TER ↓ | 114.41 | 97.82 | 101.31 | **96.94** |
| **Social Media** | | | | |
| BLEU ↑ | 28.84 | **40.43** | 29.92 | 37.52 |
| RougeL ↑ | 57.43 | **62.72** | 54.65 | 60.61 |
| TER ↓ | 206.55 | **62.66** | 216.00 | 174.00 |
| **Poetry** | | | | |
| BLEU ↑ | 18.14 | **27.07** | 12.78 | 18.37 |
| RougeL ↑ | 39.44 | **46.11** | 40.81 | 46.07 |
| TER ↓ | **108.42** | **116** | 161.55 | 121.38 |
| **Dialogues** | | | | |
| BLEU ↑ | 21.79 | 24.49 | 19.97 | **29.27** |
| RougeL ↑ | 48.69 | 54.43 | 51.46 | **55.35** |
| TER ↓ | 242.14 | **229.08** | 358.46 | **229.08** |
| **News** | | | | |
| BLEU ↑ | 10.22 | **16.17** | 8.60 | 15.42 |
| RougeL ↑ | 43.57 | 50.27 | 40.49 | **50.36** |
| TER ↓ | 77.00 | **72.12** | 120.00 | 74.09 |
| **TOTAL** | | | | |
| BLEU ↑ | 23.38 | **29.91** | 22.56 | 28.72 |
| RougeL ↑ | 49.74 | 55.47 | 50.31 | **56.03** |
| TER ↓ | 149.70 | **115.54** | 191.46 | 139.10 |

Table 3: Results.

worse than what was initially thought, and in reality, our rule-based model had the best *TER* score in the poetry domain.

It is not possible to fully determine the best-performing model. However, our rule-based model and the fine-tuned M2M stand out. Within these systems, it is clear that the performance depends on the type of sentences to be translated. Specifically, M2M achieved higher results in simpler sentences and in the domain of dialogues, whereas RB excels in the domains of social media and poetry. From this, we can infer that the M2M was able to better generalize for unseen data for simpler sentence structures. On the other hand, the rules extracted from the corpus proved to be more effective for contemporary and poetry sentence styles, aligning with the informal and formal discourse present in the videos within the corpus.

Analyzing the translations from both the rule-based and the M2M models and comparing them with the reference translation, we perceived that the majority of the errors are due to:

- **Words misplaced:** the predicted translation

includes the words present in the reference translation but in the wrong order;

- **Addition and removal of personal pronouns:** some predicted translations wrongly add/remove personal pronouns;

- **Proper nouns not correctly identified:** as proper nouns are fingerspelled in LGP, it is crucial for the models to identify them, which does not always occur.

- **Mistakes when dealing with the female gender:** some female nouns have their own sign and some are translated into MULHER (WOMAN) + MASCULINE SIGN. Some translations fail to identify the appropriate method to be followed.

In Appendix A, Table 5 demonstrates instances where our rule-based system outperforms the fine-tuned M2M model, while Table 6 showcases the opposite. Additionally, Table 7 in Appendix A displays sentences where both models yield identical outputs.

## 6 Conclusion and Future Work

We contribute with a fully automatic rule-based approach to translate EP to LGP and also with two neural models. We also contribute with an automatically labeled dataset (37,500 pairs EP/LGP) with texts from 5 different domains and a gold collection of 200 sentences (40 from each style). Our rule-based system and the neural M2M model share the podium in all scenarios. We benchmark, in this way, the task of EP to LGP translation.

For future work, different methods of evaluation should be used to measure fluency and adequacy. To better validate the results, the gold collection should be extended.

**Ethics statement**: In order to uphold the quality of our work for the end-users, that is deaf people speaking LGP, the gold set was created by LGP experts, who received certification for their work. Furthermore, both the data and code utilized in our research are accessible, in order to promote transparency and reproducibility.

## Limitations

We identify the following limitations of this work:

- The gold set should be extended with more sentences for each domain;

- A more detailed error analysis should be conducted, analyzing the idiosyncrasies of each domain;

- We should test the understandability of the sequence of glosses, even if they are not in the correct order; that is, we should test which errors are critical and which are not.

## Acknowledgements

We would like to thank the annotators Ana Sofia Fernandes and Ema Marques for generously dedicating their valuable time and expertise to assist in the development of the gold collection used and provided in this paper.

This research was supported by the Portuguese Recovery and Resilience Plan through the project C645008882-00000055 (Center for Responsible AI), and through *Fundação para a Ciência e a Tecnologia* (FCT), specifically through the INESC-ID multi-annual funding with reference UIDB/50021/2020.

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

USA. Association for Machine Translation in the
Americas.

Thomas Wolf, Lysandre Debut, Victor Sanh, Julien
Chaumond, Clement Delangue, Anthony Moi, Pier-
ric Cistac, Tim Rault, Remi Louf, Morgan Funtow-
icz, Joe Davison, Sam Shleifer, Patrick von Platen,
Clara Ma, Yacine Jernite, Julien Plu, Canwen Xu,
Teven Le Scao, Sylvain Gugger, Mariama Drame,
Quentin Lhoest, and Alexander Rush. 2020. Trans-
formers: State-of-the-art natural language processing.
In *Proceedings of the 2020 Conference on Empirical
Methods in Natural Language Processing: System
Demonstrations*, pages 38–45, Online. Association
for Computational Linguistics.

Liwei Zhao, Karin Kipper, William Schuler, Christian
Vogler, Norman Badler, and Martha Palmer. 2000. A
machine translation system from English to Amer-
ican Sign Language. In *Proceedings of the Fourth
Conference of the Association for Machine Transla-
tion in the Americas: Technical Papers*, pages 54–67,
Cuernavaca, Mexico. Springer.

# A Appendix A

| EP sentences | LGP sentences | Domain |
|---|---|---|
| Preciso ir dormir. | EU PRECISAR DORMIR | Simple Sentences |
| (I need to sleep.) | (I NEED SLEEP) | Simple Sentences |
| Queres uma xícara de café? | {XÍCARA CAFÉ TU QUERER}(q) | Simple Sentences |
| (Do you want a cup of coffee?) | ({CUP COFFEE YOU WANT}(q) | Simple Sentences |
| Vc não é uma minoria | VC MINORIA (NÃO) | Social Media |
| (You are not a minority) | (YOU MINORITY (NO)) | Social Media |
| Até questionei a minha sanidade | ATÉ SANIDADE MEU EU QUESTIONAR | Social Media |
| (I even questioned my sanity) | (EVEN SANITY MY I QUESTION) | Social Media |
| Cumpres a tua vida? | {VIDA TEU TU CUMPRIR}(q) | Literature |
| (Do you fulfill your life?) | ({LIFE YOUR YOU FULFILL}(q)) | Literature |
| E afaga o pequeno monte | PEQUENO MONTE ELE AFAGAR | Literature |
| (And strokes the small mound) | SMALL MOUND HE STROKE | Literature |
| sim, minha senhora. | SIM MEU MULHER SENHOR | Dialogue |
| (Yes, my lady.) | (YES MY WOMAN SIR) | Dialogue |
| acho triste a rapariga. | TRISTE MULHER RAPAZ EU ACHAR | Dialogue |
| (I think the girl is sad.) | SAD WOMAN BOY I THINK | Dialogue |
| Atingiu as 192 partilhas | 192 PARTILHA ELE ATINGIR | News |
| It reached 192 shares | 192 SHARE IT REACH | News |
| Se haveria alternativa aos cortes salariais | SALARIAIS ALTERNATIVA CORTES HAVER | News |
| (If there was an alternative to wage cuts) | (WAGE ALTERNATIVE CUTS THERE IS) | News |

Table 4: Examples of parallel sentences.

| RB | M2M | Reference |
|---|---|---|
| NÓS COMER TUDO ISSO ACREDITAR CONSEGUIR (NÃO) | COMER TUDO ISSO ACREDITAR EU CONSEGUIR (NÃO) | NÓS COMER TUDO ISSO EU ACREDITAR CONSEGUIR (NÃO) |
| (WE EAT EVERYTHING THAT BELIEVE CAN (NO)) | (EAT EVERYTHING THAT BELIEVE I CAN (NO)) | (WE EAT EVERYTHING THAT I BELIEVE CAN (NO)) |
| TAL DEUS MEU PERMITIR (NÃO) | TAL DEUS MEU TU PERMITIR (NÃO) | MEU DEUS TAL PERMITIR (NÃO) |
| (SUCH GOD MY ALLOW (NO)) | (SUCH GOD MY YOU ALLOW (NO)) | (MY GOD SUCH ALLOW (NO)) |
| ONDE DT(T-O-M) EU MORAR LEMBRAR CONSEGUIR (NÃO) | ONDE DT(T-O-M) EU LEMBRAR MORAR CONSEGUIR (NÃO) | DT(T-O-M) ONDE MORAR EU LEMBRAR CONSEGUIR (NÃO) |
| (WHERE DT(T-O-M) I LIVE REMEMBER CAN (NO)) | (WHERE DT(T-O-M) I REMEMBER LIVE CAN (NO)) | (DT(T-O-M) WHERE LIVE I REMEMBER CAN (NO)) |

Table 5: Example sentences where our rule-based model performs better than the fine-tuned M2M model when compared with the reference translation.

| RB | M2M | Reference |
|---|---|---|
| MEIO CAMINHO MULHER AVÔ MEU | MEIO CAMINHO AVÓ MEU | MEIO CAMINHO AVÓ MEU |
| (HALF WAY WOMAN GRANDFATHER MY) | (HALF WAY GRANDMOTHER MY) | (HALF WAY GRANDMOTHER MY) |
| {NOVO PREPARAR ACORDO OSLO}(q) | {NOVO ACORDO OSLO PREPARAR}(q) | {NÓS NOVO ACORDO OSLO PREPARADO}(q) |
| ({NEW PREPARE AGREEMENT OSLO}(q)) | ({NEW AGREEMENT OSLO PREPARE}(q)) | ({WE NEW AGREEMENT OSLO PREPARED}(q)) |
| FILME EMOÇÃO RUA POVO ENTERRO DT(T-A-N-C-R-E-D-O) MOSTRAR | FILME EMOÇÃO POVO RUA ENTERRO DT(T-A-N-C-R-E-D-O) MOSTRAR | FILME EMOÇÃO POVO RUA ENTERRO DT(T-A-N-C-R-E-D-O) MOSTRAR |
| (MOVIE EMOTION STREET PEOPLE BURIAL DT(T-A-N-C-R-E-D-O) SHOW) | (MOVIE EMOTION PEOPLE STREET BURIAL DT(T-A-N-C-R-E-D-O) SHOW) | (MOVIE EMOTION PEOPLE STREET BURIAL DT(T-A-N-C-R-E-DO) SHOW) |

Table 6: Example sentences where the fine-tuned M2M model performs better than our rule-based model when compared with the reference translation.

| RB/M2M | Reference |
|---|---|
| VOCÊ EU AJUDAR | VOCÊ AJUDAR ME |
| (YOU I HELP) | (YOU HELP ME) |
| FOLHA SPOTIFY ELEIÇÃO CHAPA LANÇAR | FOLHA SPOTIFY ELEIÇÃO CHAPA LANÇAR |
| (FOLHA SPOTIFY ELECTION CHAPA LAUNCH) | (FOLHA SPOTIFY ELECTION CHAPA LAUNCH) |
| PAÍS BENDEGÓ | NÓS PAÍS BENDEGÓ |
| (COUNTRY BENDEGÓ) | (WE COUNTRY BENDEGÓ) |

Table 7: Example sentences where our rule-based model and the fine-tuned M2M produce the same output, whether it is correct or not.