# OpenReview forum: "Enhancing Accessible Communication: from European Portuguese to Portuguese Sign Language"
_EMNLP/2023/Conference — EMNLP 2023 Findings_

### Official Review · Reviewer_9TWC · 2023-07-28

**Soundness:** 3

**Excitement:**

3: Ambivalent: It has merits (e.g., it reports state-of-the-art results, the idea is nice), but there are key weaknesses (e.g., it describes incremental work), and it can significantly benefit from another round of revision. However, I won't object to accepting it if my co-reviewers champion it.

**Missing References:**

n/a

**Paper Topic And Main Contributions:**

This paper presents a study on automatic translation between European Portuguese (EP) and Portuguese Sign Language (LGP).

The authors developed a fully automatic rule-based system that, given a parallel EP-LGP corpus, with syntactically annotated EP, extracts syntactic and morphosyntatic rule that translate syntactically annotated EP into LGP sentences (sequences of glosses, with some annotation for spelling-out words, signaling interrogatives, etc).
This rule-based system is used to automatically create a parallel EP-LGP corpus with 35k sentences, which is used to train two NMT systems (a mBART model and a M2M model).
Evaluation of the rule-based and neural systems is performed on a held-out gold corpus of 200 sentences that has been manually validated by LGP experts.

**Questions For The Authors:**

- For the sake of the paper, you *must* explain in more detail how the rules are automatically extracted. This process has to involve some sort of word aligment procedure between EP and LGP in order to extract the bilingual dictionary, right? How are the syntactic rules built? Do you have a set of hand-crafted templates?
- It would strengthen the paper and make it more attractive to a general non-Portuguese audience if it became clear that the rule extraction mechanism is sufficiently general to be applied to other languages, as long as they can provide a suitably annotated text-sign parallel corpus. Can you say something about this?
- As you state in 106-108, COLIN consists of 113 hours, only 20 of which are annotated. From those 20 hours, you were only able to use the 45 minutes that "had all the needed syntactic annotations". Can you briefly point out what you need and what's the annotation in the remaning 19h15?
- It would be nice to have some discussion on how the rule-based system behaves if its input comes from automatic parsing, and not from what I presume are gold parses. In "actual use", a text will be automatically parsed before translation, without the parses being manually corrected.
- I don't know what ELAN is (line 108). Either briefly explain what it is, or leave it out, as it isn't that relevant to know what is the particular annotation formalism you're using, only that it concerns syntax.
- Example (3): What happens to ADJ1 and ADJ2? Where does N2 come from? From the example sentence you provide, the rule has to involve more transformations than what at first seems.
- What is the TER metric? I'm assuming it's some sort of word error rate, but it would help to have a short sentence just quickly stating which metrics you're using.
- DT() indicates that the word is to be spelled out, right? Isn't this redundant with the word being represented with characters separated by hyphens, e.g. "A-N-A", or is there a reason for the two notations?
- Table 4, third example: Is "Vc" (short-form for "Você") really translated into "VC" in LGP, implying that there a different gesture than that for "VOCÊ"? Or is "VC" spelled out. In that case, shouldn't it be "DT(V-C)"?

[these questions were mostly addressed in the rebuttal]

**Reasons To Accept:**

- Addresses a rarely seen but important task.
- The rule extraction system might work for any text-sign parallel corpus (with "sign" understood as being glosses), as long as the text is syntactically annotated, which makes it more general than what is covered by this limited study.
- Is enough content for a short paper.

**Reasons To Reject:**

- There's very, very little information on how the fully automatic rule-based system works. While it's true that this is a short paper, the automatic rule extraction is the *most important* part of the paper in my view, as it enables all the rest, so not properly explaining how it works is a major weakness.

[even with the rebuttal, much is still unclear regarding the workings of the rule-extraction procedure]

**Reproducibility:**

2: Would be hard pressed to reproduce the results. The contribution depends on data that are simply not available outside the author's institution or consortium; not enough details are provided.

**Reviewer Confidence:**

3: Pretty sure, but there's a chance I missed something. Although I have a good feel for this area in general, I did not carefully check the paper's details, e.g., the math, experimental design, or novelty.

**Typos Grammar Style And Presentation Improvements:**

- What's the purpose of the preliminary experiment you show at the start of Section 5 in Table 2? You've just ended Section 4 by describing the gold corpus you've created, but then start the evaluation with a totally different corpus, not introduced before, before moving on to the evaluation over your gold corpus (which is the one we're interested in). I'd leave this preliminary experiment out of the paper and use the space you gain to better explain the automatic rule extraction.
- 030: "attempts for developing" -> "attempts at developing"
- 056: "task in hand" -> "task at hand"
- 094: "afterward" -> "afterwards"
- 100: Add a linebreak as you change topic and start describing PE2LGP.
- 180: "Portuguese-to-Portuguese" -> Perhaps "Portuguese-to-LGP" instead?
- The paper would benefit from some reformating of the examples in the various tables, for the sake of readibility, as the all-caps characters and fully justified paragraphs aren't easy to read. Perhaps using a smaller font, small caps, or something else.
- Table 7: Since you're showing cases where RB and M2M match, you don't really need three columns. Two are enough: one for RB/M2M and one for Reference.

---

> ### Author Rebuttal · Authors · 2023-08-28
>
> First of all, thank you for your feedback.
>
> We will now proceed to answer your questions:
> - Questions regarding the rule-based system:
>      - **Q1:** The process to automatically extract the rules starts by extracting the Portuguese sentences, the LGP sentences, and the grammatical information such as part of speech tags, subjects, and objects from ELAN. The next step is to analyze the Portuguese sentences to obtain their grammatical information. Having the information regarding both languages, the alignment between the Portuguese words and the LGP signs is performed. This alignment is based on an algorithm of similarity measures, string matching, and semantic similarity. From the aligned word-sign pairs, the rules are created. We do not have a set of hand-crafted rules. We only have the information extracted from ELAN.
>      - **Q2:** The rule extraction mechanism is sufficiently general to be applied to other languages as long as a suitably annotated text-sign parallel corpus is provided.
>      - **Q3:** From the 20 hours, only 45 minutes have all the needed information to create the rules. This information consists of the Portuguese sentences, the LGP glosses, the type of sentence, the part of speech tags, and the syntactic analysis. The other 19h15 still do not have all this information.
>      - **Q4:** We are not sure that we understood the question. The parsing is always automatic. Even the one that is on the basis of the gold collection. Only the gold collection received manual correction/validation by the experts (in order to be created).
> - Questions regarding clarifications:
>      - **Q5:** ELAN is a software tool that enables the annotation of audio and video data. ELAN allows users to create multiple layers of annotations, which are synchronized and aligned with the corresponding audio or video segments.
>      - **Q7:** The TER metric returns the minimum amount of operations needed to convert the predicted translation into the reference one - includes insertion, deletion, and substitution of words and shifts within the sentence. The lower the value, the better.
> - Questions regarding annotations' clarifications:
>      - **Q6:** For the Portuguese sentence:
>
>          "Há (V1) grandes (ADJ1) desenvolvimentos (N1) artísticos (ADJ2)"
>           (There are (V1) great (ADJ1) aristic (ADJ2) developments (N1)),
>
>           the corresponding sentence in LGP glosses is:
>
>           "TER-MUITO (V1) ARTE (N2) DESENVOLVIMENTO (N1)"
>           (HAVE-A-LOT (V1) ART (N2) DEVELOPMENT (N1)).
>
>           This occurs because “There are great”  is converted into only one sign “HAVE-A-LOT” which corresponds to the Verb (V1).  “artistic” is converted into “ART” which is a noun that was not present in the original sentence and is thus assigned a new number (2). Taking this into consideration, the ADJ1 and ADJ2 are removed from the sentence.
>      - **Q8:**  The annotation for the words that are fingerspelled is the annotation used by PE2LGP (defined by experts in LGP) and therefore used by us.
>      - **Q9:** The example "Vc" represents an error where the system cannot understand that it should generate the sign "VOCÊ".

---

### Official Review · Reviewer_Ue8D · 2023-08-05

**Soundness:** 2

**Excitement:**

3: Ambivalent: It has merits (e.g., it reports state-of-the-art results, the idea is nice), but there are key weaknesses (e.g., it describes incremental work), and it can significantly benefit from another round of revision. However, I won't object to accepting it if my co-reviewers champion it.

**Paper Topic And Main Contributions:**

The paper addresses the challenge of translating European Portuguese into Portuguese Sign Language. The authors create a rule-based translation system, a 5-domain parallel corpus between these languages and fine-tune two large multilingual neural machine translation models, mBART and M2M. All proposed models are evaluated on a previously available evaulation set and a new 5-domain newly annotated gold set.

The evaluation shows that the M2M-based method achieves the best overall performance on the original evaluation set while the rule-based system achieves the highest results on the new evaluation set. The qualitative evaluation shows some problems related to specific language constructs.


**Questions For The Authors:**

Why was the gold collection created in such way (pre-annotated by rule-based system, separate annotation sets)? Isn't this method correlated with better results of the ruled based system?

**Reasons To Accept:**

1. The paper addresses an important and challenging problem of translating between European Portuguese and Portuguese Sign Language.
2. The paper's approach presents several new systems and their evaluation.
3. The paper introduces the new dataset, comprising sentences from diverse domains.

**Reasons To Reject:**

1. The major problem I have with this paper is its evaluation methodology on the newly created gold set. Having a gold collection annotated in parallel on a plain (not pre-annotated) text and calculating inter-annotator agreement would make it more reliable.
2. The metrics used may not fully capture the quality of translation, especially for sign language. Additional metrics, such as fluency, adequacy, and subjective human evaluations, already started and presented in Appendix 1, could provide a more comprehensive assessment.
3. The paper briefly mentions common errors made by the models but does not go into detail regarding the root causes or potential strategies to mitigate them.
4. The paper mentions that the COLIN corpus includes signers ranging from 4 to 89 years old, but we don't know how it relates to the LGP corpus.

**Reproducibility:**

3: Could reproduce the results with some difficulty. The settings of parameters are underspecified or subjectively determined; the training/evaluation data are not widely available.

**Reviewer Confidence:**

3: Pretty sure, but there's a chance I missed something. Although I have a good feel for this area in general, I did not carefully check the paper's details, e.g., the math, experimental design, or novelty.

**Typos Grammar Style And Presentation Improvements:**

- 039: from now on COLIN – COLIN should be explained here (is in line 104); moreover, it seems there is no citation/URL of COLIN anywhere
- 053: evaluated in a gold collection > on a gold collection?
- 055: we hope to contribute to benchmark this task > we hope to contribute to benchmarking this task
- 056: To illustrate the task in hand > at hand
- in footnote 2: Developed in project > Developed in the project
- 094: afterward > afterwards
- 165 and later: please consider using dashes instead of minuses when sequences of elements in syntactic structures are enumerated
- the readability of the equations should be improved a bit (e.g. 'NP' and 'VP' should be separated a bit more than 'N' and 'P' in 'NP'; arrows should be arrows)
- 166: a LGP sentence  > an LGP sentence
- 202: 7500 times 5 domains gives 37500; why do we have 35k here?
- 262: in a wrong order > in the wrong order
- in Table 3: 49,74 > 49.74; 29.906 > 29.91
- in the bibliography: please check the case of letters to avoid lowercase names (e.g. 'european portuguese').
- please reformat all tables to conform to EMNLP 2023 style!
- why aren't the tables in the appendix on the same page as its heading?

In Figures:
- Figure 1 has a caption-ending dot while Fig. 2 and 3 don't – please make it consistent with EMNLP 2023 guidelines
- maybe 'results' in the caption of Figure 2 should be lowercase?
- please consider right- and dot-aligning the numbers, the results may read better

---

> ### Author Rebuttal · Authors · 2023-08-28
>
> First of all, thank you for your feedback.
>
> The gold collection was previously annotated by the rule-based system in order to facilitate the annotators' job. If they agree with the results they do not change them; otherwise, they will correct them. In such approaches to annotation, there is always some risk of "bias" towards the initial annotation. However, we do not think this has affected the results.
>
> If the paper is accepted, we will use the extra page to add additional details about the evaluation methodology and metrics.

---

### Official Review · Reviewer_LV1i · 2023-08-10

**Soundness:** 3

**Excitement:**

4: Strong: This paper deepens the understanding of some phenomenon or lowers the barriers to an existing research direction.

**Missing References:**

Cite BLEU (whichever method was used), TER and ROUGE papers!

On historical progress in SLMT:
"A survey on Sign Language machine translation" (Núñez-Marcos et al., 2023)

T2G and G2T in the neural era:
"Sign Language Transformers: Joint end-to-end Sign Language Recognition and Translation" (Camgöz et al., 2018)
"Approaching Sign Language Gloss Translation as a Low Resource Machine Translation Task" (Zhang and Duh, 2021)

A different hybrid RB/NMT approach:
"Translating Spanish into Spanish Sign Language: Combining Rules and Data-driven approaches" (Chiruzzo et al., 2022)

Standardisation of SL datasets and what is annotated:
"Digging into Signs: Emerging Annotation Standards for Sign Language Corpora" (Cormier et al., 2016)
"Universal dependencies for Swedish Sign Language" (Östling et al., 2017)

**Paper Topic And Main Contributions:**

This short paper introduces new parallel resources for Portuguese Sign Language (which is an extremely low resource language) and Portuguese (Portugal): These include a considerable corpus as well as an expert-annotated gold-standard test set which could provide an evaluation benchmark for other work on LGP.

It also uses these resources to conduct experiments in Sign Language Machine Translation (SLMT), restricted to the text-to-sign language gloss direction. Two transformer-based neural models are tuned for the task, as well as a rule-based (RB) model which automates a process invented in a previous work (Gonçalves et al., 2021). The authors also compare this previous RB model on the same testing data in order to provide comparability between studies against a number of metrics.

The authors also select some qualitative output from each model to be shown in an Appendix.

It is envisaged that the dataset and code used to generate and evaluate the results will become available after the period of anonymity in order to allow reproducibility and verification of results/method.



**Questions For The Authors:**

Question A: Did the authors consider pre-processing glosses for either lexicalisation or standardisation of what is annotated? (Related references shown below, related to text around line 55)

Question B: How many individual signers are included in the COLIN dataset?

Question C: What concrete future work is planned, and how will these models improve accessibility and communication to the LGP community?

In order to have the required specificity for reproducibility, the following shorter questions ask the following:

Question D: what manual tasks are needed to create PE2LGP (line 43)?

Question E: What linguistic information exactly is extracted from the corpus (line 145)?

Question F: What exactly were the computational resources available, were there GPUs (line 194)?

Question G: Which methods of calculating BLEU and ROUGE are used (and why)(should be described in Section 4)?

Question H: How do you interpret the results of the evaluation metrics, and how do they link to your qualitative results (section 5)?

Question I: How were sentences picked for the gold set (section 4.3)? Were they cherry-picked or chosen at random? This could create bias if they were selected with a certain motive in mind, and affect future testing of SLMT models

**Reasons To Accept:**

This paper presents a large body of data including a small gold-standard, expert annotated test set to facilitate future work on machine translation and computational linguistic studies into LGP. This is of benefit to the CL and NLP community, especially as LGP is a particularly low resource language and on behalf of a historically marginalised community.

The methodology is mostly described well - and though missing some specifics in the formulation of rules, model calibration, and evaluation metrics - it should be relatively straightforward to attempt to reproduce the findings in this paper. This is also in part due to the commitment of the authors to release data and code as open source.

The methodology itself is also robust and suitable for the resources available and novelty of the task. A rule-based approach is a relevant next step given the work it builds upon (in both LGP and San-Segundo's approach in LSE), and adding neural/hybrid experiment shows a commitment to test recent innovations on this data.

The authors also comment on the limitations of the approach taken and contents of the data, and it is hoped that future research will tackle these.

In my opinion, this short paper is suitable for publication with the integration of some of the suggestions and answering of some of the questions in this review.

**Reasons To Reject:**

Overall, the layout of the paper and its contents are adequate. However, the most serious flaws are found in the experimental results sections.

Evaluation metrics are briefly referred to in the text and are included in the Table, but they are not introduced or described in the text. It is necessary to know for example which type of BLEU (BLEU-4/sacreBLEU?/ROUGE-F1? see also references section) are used, and a more clear description of what higher/lower scores may mean or imply. The description of these results can be rather shallow in that certain settings and metrics are described as "better" than others. There is no further explanation or theoretical discussion about what these results imply and what could be done to mitigate them.

My main suggestion for improvement in this regard is to use the extra page allowed post-review to focus on being more specific in dataset, model parameter, and evaluation metric description. In addition, a lengthier discussion of results would ground this study in its research context in a helpful manner for the reader.

Another small comment is that the SoTA section (lines 122-136) does not mention many key works in the history of SLMT. A greater discussion of relevant technologies - such as Text2Gloss, RB vs. statistical vs. neural, or E2E vs. modular SLMT - where it references the present study would be of great benefit.

**Reproducibility:**

3: Could reproduce the results with some difficulty. The settings of parameters are underspecified or subjectively determined; the training/evaluation data are not widely available.

**Reviewer Confidence:**

4: Quite sure. I tried to check the important points carefully. It's unlikely, though conceivable, that I missed something that should affect my ratings.

**Typos Grammar Style And Presentation Improvements:**

It should maybe be made more clear in the title and key summary sections that this is text-to-SL Gloss translation (and this unidirectional translation may be a limitation of use). This is to compare it with bidirectional systems, but also with E2E systems such as text->SL avatar or SL video->text which have very different implications in the real-world applications of this technology.

Perhaps refer to European Portuguese by the more standard ISO reference "pr-PT" instead

Line 113: Automatize -> Automate

Line 141: Suggest "to reconstruct the expert translator's grammatical rules"

Numbers <= twelve to be written in text in the main body of the text

35000 -> 35,000

Line 196: Sentence could be rephrased for clarity

Line 217: (tone) "famous Portuguese poet" instead

Line 267: Use "fingerspelled" instead

Table 3: Upwards and downwards arrows next to the metric names to state whether a higher or lower figure is desirable

Table 2/3: Make sure all numbers have either an equal number of digits after the decimal point/equal number of significant figures

Table 3: Wrong TER is bolded in "News"

---

> ### Author Rebuttal · Authors · 2023-08-28
>
> First of all, thank you for your feedback.
>
> With respect to your main suggestion, and if the paper is accepted, we will use the extra page to add additional details about the dataset, model parameters, and evaluation metrics.
>
> We will now proceed to answer your questions:
> - **Question A:** We have not pre-processed the glosses.
> - **Question B:** We do not know how many individual signers are in the COLIN dataset. That information is not available.
> - **Question C:** Concrete future work is to use the current models that generate the sequence of glosses to improve an existing avatar that reproduces the signs associated with the glosses.
> - **Question D:** The manual tasks that were needed to create the PE2LGP are:
>      1. Check if the rules created are grammatically sound (for example: rules that have a Portuguese structure but do not have an LGP one should be discarded);
>      2. Check if the alignment between Portuguese words and LGP glosses makes sense;
>      3. Identify glosses that need to be fingerspelled.
> - **Question E:** The linguistic information that is extracted from the corpus is composed of: the Portuguese sentences, the LGP sentences, and all the grammatical information such as part of speech tags, subjects, and objects.
> - **Question F:** Yes, there were GPUs. Two GPUs with 50GB of memory.
> - **Question G:** For calculating BLEU and ROUGE we used the metrics from the HuggingFace library. The BLEU score used is the BLEU-4 as it is usually the most common one to evaluate translation systems. The ROUGE used is the rougeL that scores based on the longest common subsequence. We will clarify this in the paper.
> - **Question H:** After obtaining the results of the automatic evaluation metrics, we manually compared the reference and the predicted translations and concluded what were the main problems, which we briefly describe in the paper. Although not mentioned in the paper, more specific metrics might be needed to effectively evaluate the system (for example: evaluate proper nouns and female gender not correctly identified).
> - **Question I:** The sentences for the gold set were randomly picked from the 5 domains described in Section 4.1. We will clarify this in the paper.

---

### Meta-Review · Area_Chair_pSXD · 2023-09-17

**Recommendation:** 4

**Metareview:**

This short paper introduces new parallel resources for Portuguese Sign Language (which is an extremely low resource language) and Portuguese (Portugal): These include a considerable corpus as well as an expert-annotated gold-standard test set which could provide an evaluation benchmark for other work on LGP.

Reasons To Accept:
- This paper presents a large body of data including a small gold-standard
- The methodology is mostly described well
- The methodology itself is robust and suitable for the resources available and novelty of the task
- Addresses a rarely seen but important task

Reasons To Reject:
- Having a gold collection annotated in parallel on a plain (not pre-annotated) text and calculating inter-annotator agreement would make it more reliable.
- The metrics used may not fully capture the quality of translation, especially for sign language. Additional metrics, such as fluency, adequacy, and subjective human evaluations, already started and presented in Appendix 1, could provide a more comprehensive assessment.
- There's very, very little information on how the fully automatic rule-based system works. While it's true that this is a short paper, the automatic rule extraction is the most important part of the paper in my view, as it enables all the rest, so not properly explaining how it works is a major weakness.


In my humble opinion, this work is very important to develop tools for sign language. The golden dataset is very precious and this gift to the community is really significant.

---

### Decision · Program_Chairs · 2023-10-07

**Decision:**

Accept-Findings

**Comment:**

This short paper introduces new parallel resources for Portuguese Sign Language (which is an extremely low resource language) and Portuguese (Portugal): These include a considerable corpus as well as an expert-annotated gold-standard test set which could provide an evaluation benchmark for other work on LGP.

Reasons To Accept:
- This paper presents a large body of data including a small gold-standard
- The methodology is mostly described well
- The methodology itself is robust and suitable for the resources available and novelty of the task
- Addresses a rarely seen but important task

Reasons To Reject:
- Having a gold collection annotated in parallel on a plain (not pre-annotated) text and calculating inter-annotator agreement would make it more reliable.
- The metrics used may not fully capture the quality of translation, especially for sign language. Additional metrics, such as fluency, adequacy, and subjective human evaluations, already started and presented in Appendix 1, could provide a more comprehensive assessment.
- There's very, very little information on how the fully automatic rule-based system works. While it's true that this is a short paper, the automatic rule extraction is the most important part of the paper in my view, as it enables all the rest, so not properly explaining how it works is a major weakness.


In my humble opinion, this work is very important to develop tools for sign language. The golden dataset is very precious and this gift to the community is really significant.